# Study on Properties and Degradation Behavior of Poly (Adipic Acid/Butylene Terephthalate-Co-Glycolic Acid) Copolyester Synthesized by Quaternary Copolymerization

**DOI:** 10.3390/ijms24076451

**Published:** 2023-03-29

**Authors:** Yanning Wang, Boyou Hou, Liping Huang, Bingjian Li, Shi Liu, Mingyang He, Qun Chen, Jinchun Li

**Affiliations:** 1School of Materials Science and Engineering, Changzhou University, Changzhou 213164, China; 2Jiangsu Key Laboratory of Advanced Catalytic Materials and Technology, Changzhou University, Changzhou 213164, China; 3Jiangsu Key Laboratory of Environmentally Friendly Polymeric Materials, National-Local Joint Engineering Research Center of Biomass Refining and High-Quality Utilization, Changzhou University, Changzhou 213164, China

**Keywords:** methyl glycoacetate, glycolic acid, biodegradable, hydrolysis

## Abstract

At present, the development and usage of degradable plastics instead of traditional plastics is an effective way to solve the pollution of marine microplastics. Poly (butylene adipate-co-terephthalate) (PBAT) is known as one of the most promising biodegradable materials. Nevertheless, the degradation rate of PBAT in water environment is slow. In this work, we successfully prepared four kinds of high molecular weight polyester copolyesters (PBATGA) via quaternary copolymerization. The results showed that the intrinsic viscosity of PBATGA copolymers ranged from 0.74 to 1.01 dL/g with a glycolic acid content of 0–40%. PBATGA copolymers had excellent flexibility and thermal stability. The tensile strength was 5~40 MPa, the elongation at break was greater than 460%, especially the elongation at break of PBATGA10 at 1235%, and the thermal decomposition temperature of PBATGA copolyesters was higher than 375 °C. It was found that PBATGA copolyester had a faster hydrolysis rate than PBAT, and the weight loss of PBATGA copolymers showed a tendency of pH = 12 > Lipase ≈ pH = 7 > pH = 2. The quaternary polymerization of PBAT will have the advantage of achieving industrialization, unlike the previous polymerization process. In addition, the polymerization of PBATGA copolyesters not only utilizes the by-products of the coal chemical industry, but also it can be promising in the production of biodegradable packaging to reduce marine plastic pollution.

## 1. Introduction

Plastic pollution is a global environmental problem [1,2,3,4]. Some studies predicted that about 12,000 tons of plastic waste would be buried in landfills or the natural environment by 2050 [5]. The global plastic production may increase by two times by 2100 [6], among which the untreated plastic in the natural environment will eventually be brought to the marine and river water environment through water circulation. According to statistics, about 768 million tons of plastic waste eventually enter our oceans every year [7], resulting in more and more serious marine plastic pollution and serious harm to marine organisms.

Currently, one of the strategies that can effectively solve plastic pollution is to replace most traditional disposable non-degradable plastics with biodegradable plastics that have been put into the market [8,9,10]. However, the degradation rate of these biodegradable plastics is slow in the ocean. The biodegradable plastics have little effect on the treatment of marine plastic pollution [11,12,13].

Poly (butylene adipate-co-terephthalate) (PBAT) is one of the most widely used degradable materials in the market in the research of biodegradable plastics owing to its super-high toughness while being a 100% compostable polyester [14,15]. It is mainly used in packaging products [16] and agricultural film [17,18,19]. However, the degradation of PBAT is slow in water environments [20]. Therefore, it is of great significance to improve the hydrolysis rate of PBAT. A few researchers have studied the degradation of PBAT in water environments. Y-X. Weng et al. [21] studied the hydrolysis behavior of PBAT/PLA film in real soil. The study showed that the melting point of PBAT increased from 120 °C to 124 °C after 4 months of degradation. G.X. De Hoe et al. [22] confirmed that adding a light stabilizer to PBAT film can reduce the negative impact of ultraviolet radiation on the hydrolysis ability of PBAT enzyme. M.T. Zumstein et al. [23] explored deeply the weight change and dissipation dynamics of the PBS membrane in the hydrolysis process at the molecular level by using quartz crystal microbalance technology (QCM-D) as the main analytical technology. C. Shen et al. [24] found that PBAT could be effectively degraded by using a small amount of alkali in a solution of ethanol and water. The degradation rate of PBAT reached more than 99% after the samples were kept at 80 °C for 50 min, and the high concentration of ethanol solution was conducive to the separation of hydrolysates, realizing the complete recovery of the PBAT monomer under mild conditions. T. Kijchavengkul et al. [25] studied the hydrolysis behavior of PBAT film in phosphate buffer solution with pH = 8.0 at 58 °C, and measured the effect of hydrolysis on each comonomer in PBAT by using the change of the mole fraction of BA unit and BT unit. The results showed that the ester group in aliphatic BA unit was easier to hydrolyze than in the aromatic BT unit. In addition, PBAT has a faster degradation in manure compost than yard compost or food waste compost. In the process of biodegradation, the increasing crystallinity of PBAT is observed, which indicated that the biodegradation rate of the amorphous region is faster than that of the crystalline region. Q. Deshoules et al. [26] studied the degradation behavior of PBAT (0.3 mm in thickness) in deionized water at 80~100 °C for 14 days. The study showed that when the average molecular weight (M_n_) of PBAT decreased to 11,000 g/mol, PBAT changed from a ductile material to a brittle material, which is the starting point of the formation of PBAT microplastics. Sangroniz et al. [27] studied the degradation behavior of PBAT/bisphenol A hydroxyether (PH) composites (0.2 mm in thickness) in water and showed that the mass loss of PBAT was about 50% after 405 days in water at 60 °C, while the mass loss of PBAT-modified materials containing 25% and 50% PH was about 35% and 20%, respectively. H. Hu et al. [28] introduced a diglycolic acid monomer with a more hydrophilic structure into PBAT to obtain poly(butylene adipate- co-diglycolate-co-terephthalate) (PBADT) copolyesters by melt polycondensation, and investigated the degradation behavior of PBADT polyesters with different copolymerization ratios in PBS buffer solution at 37 °C. It was found that the mass loss of PBAT was only 3.6% after 42 days of placement, while the mass loss of PBA_30_D_20_T (the molar content of butylene adipate was 30%, the molar content of butylene diglycolate was 20%) and PBA_0_D_50_T copolyesters was 5.8% and 16.3%, respectively, with the addition of diglycolic acid, indicating that the PBADT copolyesters were more rapidly hydrolyzed. Subsequently, Y. Dong et al. [29] synthesized poly(butylene diglycol ester-furandicarboxylate) (PBDF) copolyesters with more hydrophilic as well as barrier properties than PBAT by ester exchange reaction and melt polycondensation, with a 46% mass loss of PBDF40 after 28 days of CALB enzymatic hydrolysis. Y. Ding et al. [30] introduced glycolic acid (GA) segments into a polybutylene succinate (PBS) main unit through ternary copolymerization, and poly (butylene succinate-co-glycolate) (PBSGA) copolymer with GA contents of 0~40% were successfully obtained. The degradation performance of the PBSGA copolyester in five different solutions was studied by measuring the weight loss. The results showed that the degradation rate of PBSGA copolymer in these five solutions was in the order of alkaline > Candida antarctica lipase B (concentration was 0.1 mg/mL) > acidic > neutral > NaCl salt solution.

In our previous work, we successfully synthesized a series of poly (adipic acid/butylene terephthalate-co-glycolic acid) copolyesters (PBATGA) with high molecular weight, taking methyl glycoacetate as the raw material [31]. However, the polymerization process is complex because the oligomer methyl glycoacetate (OMG) needs to be synthesized in the first step. Therefore, in order to simplify the polymerization process and reduce the operation cost, and further evaluate the degradation performance of PBATGA copolyester, four kinds of PBATGA copolyesters were prepared using only a one-step polymerization process based on methyl glycoacetate, 1,4-butanediol, adipic acid and terephthalic acid as monomers according to different molar ratios.

## 2. Results and Discussion

### 2.1. GPC and ^1^HNMR Analysis of PBATGAs

Table 1 shows the characteristic viscosity ([η]), numerical average molecular weight (Mn), heavy average molecular weight (Mw), polymer dispersity index (PDI) and molar content of GA for PBAT and PBATGA copolyesters (n_GA_), and the results showed that when the molar ratio of PTA + AA/GA is 0~1.5, the average molecular weight of the obtained polymers ranged from 20,663 to 31,084 g/moL, and the corresponding characteristic viscosity ranged from 0.74 to 1.01 dL/g. Meanwhile, the molecular weight distribution of the copolyester became narrower with the addition of GA, with a range of values from 1.40 to 1.63. Furthermore, it was found that the viscosity of PBAT and PBATGA copolyesters were between poly(butylene adipate-co-butylene furandicarboxylate) (PBAF) and poly (ethylene aliphatate-co-terephthalate) (PEAT) through comparison. The molecular weight of PBATGA copolyester obtained by the one-step method is lower than that obtained by the two-step method. However, the advantages of the synthesis process included high yield, short process and simple process.

Figure 1a,b shows the nuclear magnetic resonance (NMR) hydrogen spectra of PBATGAs, and the average molar content of GA units in the copolymer was calculated using Equation (5). The chemical shift at δ = 8.06 ppm (a) was attributed to the hydrogen proton peak on the benzene ring, the peaks at δ = 4.40 ppm (c), 4.34 ppm (d) and δ = 1.95 ppm (h) indicated the peaks of BDO in the BT unit. Additionally, the peaks at δ = 4.11 ppm (e) and δ = 4.05 ppm (f) belongs to −OCH_2_− of BDO in the BA unit, δ = 2.30 ppm (g) represented the proton peaks of AA in the BA unit, and the methylene peak in the GA unit was at δ = 4.6~5.0 (b_1_–b_5_) ppm. The intermediate methylene proton peaks (e_1_–e_3_) of BDO were also shifted when BDO was attached to GA. Similarly, when AA was attached to BDO and GA, the new peak g1 appeared at δ = 2.41 ppm. The shift of the proton peak in BDO and the appearance of the new peak in AA indicated that the GA unit was successfully attached to the molecular backbone, Thus, PBATGA copolymers could be successfully synthesized by quaternary copolymerization.

Overall, n_GA_ could be calculated by Equation (1),
(1)$$n_{GA}=\frac{2I_{b}}{I_{a}+2I_{b}}$$

### 2.2. DSC Analysis

The DSC curves of PBATGAs during cooling and secondary heating are displayed in Figure 2a,b, and their characteristic parameters are summarized in Table 2. As shown in Figure 2a, PBAT and PBATGA10~30 typically observed crystallization peaks, and PBATGA40 showed only slight crystallization. The crystallization temperature (T_C_) and crystal enthalpy (ΔH_c_) of PBAT are 75.23 °C and 24.55 J/g, respectfully, and the crystallinity (X_c_) is 21.54%. T_C_ and X_c_ of PBATGAs decreased with increasing GA content; T_C_ decreased from 75.23 °C to 27.51 °C and X_C_ decreased from 21.55% to 7.95%. This is mainly ascribed to the presence of GA units disturbing the regularity and symmetry of the PBAT main chain, resulting in a decrease in the crystallization capacity of PBATGA copolyester. Meanwhile, the crystallinity and lamellar thickness of the PBATGA crystal are reduced, leading to a decrease in the melting point. As shown in Figure 2b, the melting temperature (T_m_) and melting enthalpy (ΔH_m_) of PBAT were 130.5 °C and 16.13 J/g, respectively. The melting point of PBATGA10–40 decreased from 133 °C to 100.6 °C with increasing content of GA, and only PBATGA40 showed obvious cold crystallization, indicating that the crystallization rate of PBATGA40 was very slow under this condition, and PBATGA40 was more possibly an amorphous copolyester. However, PBATGA10–30 possessed good crystallization performance. In comparison with PBAF and PEAT, PBATGA10–30 seems to possessed a good crystallizability.

The crystallinities (X_c_) were calculated according to Equation (2):(2)$$X_{c}\;(\%)=\frac{{\Delta H}_{m}}{f*{\Delta H}_{m}^{\infty }}\;\times \;100\%$$
where ΔH_m_ is the measured heat of fusion, *f* is the weight fraction of PBAT and ΔH*_m_*^∞^ is the enthalpy fusion for a crystal having infinite crystal thickness (114 J/g for PBAT) [34].

### 2.3. TG Analysis

Figure 3a showed the thermal decomposition curves for PBATGA copolymers and Figure 3b showed the DTG–T curves for PBATGA copolymers. The temperatures corresponding to the 5% weight loss (T_5%_) and the maximum decomposition temperatures (T_d, max_) of PBATGA copolymers are summarized in Table 2. As shown in Figure 3, the decomposition temperatures of PBATGA copolymers decreased with increasing molar content of GA. The temperature of PBATGA copolymers at 5% weight loss dropped from 382.0 °C to 375.2 °C when the GA content was 40%. Though the introduction of GA units slightly weakened the thermal stabilities of PBATGA copolymers, they still have good thermal stabilities, and the decomposition temperature is higher than the melting temperature. Therefore, it could be processed by injection molding, extrusion molding, blow molding and other processing processes below 300 °C. In conclusion, the PBATGA copolyesters have excellent thermal stability in this work.

### 2.4. DMA Analysis

PBAT was a semi-crystalline polymer, as evidenced by the DSC figure above. In order to study the effect of GA groups on the glass transition temperatures (T_g_) of PBATGA polyesters, a series of PBATGA copolymers were investigated using DMA. The loss factor (tanδ) and modulus (E′) of PBATGA copolyesters with varying GA contents as a function of temperature are displayed in Figure 4. As shown in Figure 4a, the tanδ curves of the copolyester showed a relaxation process (α relaxation) in the measurement temperature range; that is, the glass transition temperature (T_g_), and the temperature corresponding to the maximum height value of the tanδ peak is the T_g_ that characterizes the movement of the molecular chain, and the relevant data are shown in Table 2. PBATGA copolyesters exhibited a single glass transition temperature, the T_g_ values of PBATGA10–40 were between −20.1 °C and −14.5 °C, which was close to the T_g_ of PBAT (−17.4 °C), possibly because their flexibility effects on PBAT segments is not obvious, while GA segments exist at the same molecular weight, resulting in little change in T_g_ value.

### 2.5. Mechanical Properties

The stress–strain curve plot of PBATGA are displayed in Figure 5, and the relevant data are summarized in Table 3. As shown in Figure 6, PBAT and PBATGA exhibited typical ductile fractures. The tensile strength and elongation at break of PBAT were 18.58 MPa and 896%, respectively. The tensile strength of PBATGA copolyester gradually decreased from 18.58 MPa to 5.89 MPa with increasing content of GA, and the elongation at break showed a trend of first increasing and then decreasing, remaining in the range of 400~1300%. This is a result of the combination of DSC; it is precisely because with increasing content of GA the crystallization properties of PBATGA decrease, resulting in the crystallinity decrease, which leads to the tensile strength decrease. It is worth noting that the tensile strength of PBATGA40 is low, mainly due to increased content of GA and the difficulty of increasing the polymerization degree. Young’s modulus and yielding stress tended to decrease with the increasing content of GA units. The trend could be attributed to the decrease of crystallinity.

In addition, it could be seen from Figure 6 there was an obvious strain−hardening stage and almost no obvious yield point from PBAT to PBATGA20. This is because during the high elastic deformation caused by the orientation of the molecular chain along the external force under the action of a small external force, the microcrystals are also rearranged. Some crystals may even break into smaller units, and then recrystallize in orientation. However, the strain−hardening stage decreases or even disappears from PBATGA30 to PBATGA40. This is mainly because with more GA units, there is less crystal area in PBATGA copolyester, which makes the polymer unable to withstand a larger load. Accordingly, the presence of GA reduced the strain−hardening stage in the stress–strain curve.

Compared with other biodegradable polyesters (Table 3), PBATGA copolyesters have the best toughness and higher strength than linear low−density polyethylene (LLDPE) [35], low−density polyethylene (LDPE) [36], PBSA, PBAF, PEAT and poly(propylene carbonate) (PPC). In summary, PBATGA copolyesters have a lower modulus.

**Table 3 ijms-24-06451-t003:** Mechanical performance parameters of PBATGAs.

Sample	Tensile Strength at Break (MPa)	Elongation at Break (%)	Young’s Modulus (MPa)	Yielding Stress (MPa)
PBAT	18.6 ± 1.0	896 ± 157	43.0 ± 9.5	7.1 ± 6.5
PBATGA10	18.0 ± 0.5	1235 ± 108	28.8 ± 0.8	7.4 ± 0.5
PBATGA20	13.0 ± 1.4	1028 ± 322	26.6 ± 3.1	7.2 ± 2.4
PBATGA30	9.3 ± 0.5	922 ± 4	24.6 ± 1.8	6.6 ± 1.5
PBATGA40	5.9 ± 0.1	463 ± 22	36.0 ± 1.3	4.2 ± 0.2
PBAF50 [32]	15.3 ± 1.1	365 ± 78	41.5 ± 2.9	−
PEAT50 [33]	22.0 ± 0.4	762 ± 7	42.0 ± 2.3	−
PBS [37]	43.7 ± 2.3	561 ± 16	220.0 ± 7.5	32.7 ± 2.2
PBSA [38]	12.8 ± 2.8	21 ± 10	−	−
PPC [39]	6.6 ± 0.4	3 ± 43	−	−

### 2.6. Degradation Behaviors of PBATGA Copolymers in Different Buffer Solutions

To investigate the degradation of PBATGA copolyesters in water, PBATGA copolyesters were placed in 4 different buffer solutions (pH = 2, 7, 12, Lipase) for 49 days. The weight variation of PBATGA copolyester in four different PBS buffer solutions are shown in Figure 6. The degradation rates of copolyesters in different pH environments were different; the degradation rates of PBATGA copolyesters in these four buffers were, in descending order, pH = 12 > Lipase ≈ pH = 7 > pH = 2. For the same buffer solution, the hydrolysis rate of PBATGA copolyesters was faster as the GA content increased, indicating that the addition of GA groups accelerated the hydrolysis of PBAT, which is due to the ester bond in the GA unit that can be hydrolyzed rapidly [40]. This result ties in well with previous studies discussing copolymerization of oligomeric glycolic acid into PBS [41] and poly (butylene furandicarboxylate) (PBF) [42]. The weight loss of neat PBAT in Lipase buffer solution was 8.97% after 49 days, while the weight loss of PBATGA copolyesters gradually increased with increasing content of GA; PBATGA40 decreased the most with a weight loss of 15.27. In addition, it is obvious from Figure 6 that the PBATGA copolyesters degraded the fastest under alkaline conditions. For PBATGA40, the weight loss was 58.29% after 49 days in alkaline solution; however, the weight loss in acidic and neutral solutions was only 12.67% and 17.21%, respectively, which can be ascribed to how the water−soluble material can react with the hydroxide ions in the degradation solution during the hydrolysis process due to the presence of terminal carboxyl groups. In summary, copolymerization of methyl hydroxyethanolate monomer into the PBAT main chain can accelerate the hydrolysis of PBAT.

### 2.7. Degradation of PBATGAs in Humid Air at Room Temperature

In order to further study the degradation behavior of PBATGA copolyesters, the samples of PBATGA copolyester (1 mm in thickness) were placed in an environment with a temperature of 25 °C and a humidity of 70% for 3 months, close to that of the true environment. The molecular weight and molecular weight distribution changes during sample storage are summarized in Table 4. After the first month’s storage, the molecular weight of PBAT and PBATGA10 decreased gradually, the number average molecular weight of PBAT and PBATGA10 decreased by 3748 and 4540, respectively. However, the number average molecular weight of PBATGA20~40 reduced between 6000 and 10,000. The molecular weight of PBATGA copolyester decreased slowly after storage for 2 months. In addition, the molecular weight distribution of PBATGA copolymers gradually widened over time. These results showed that the presence of GA unit will accelerate the degradation of PBAT in an ordinary environment, and make the storage stability worse, which can be ascribed to the GA units in molecular chain, because they are susceptible to hydrolysis [43].

## 3. Materials and Methods

### 3.1. Materials

1, 4−butanediol (BDO, 99%) was purchased from Sinopharm Chemical Reagent Co., Ltd. (Shanghai, China). Adipic acid (AA, 99.5%) was obtained from Shanghai Lingfeng Chemical Reagent Co., Ltd. (Shanghai, China). Terephthalic acid (PTA, 99%) was purchased from Aladdin Chemical Co., Ltd. (Shanghai, China). Methyl glycolate (MG) was purchased from Shanghai Pujing Chemical Co., Ltd. (Shanghai, China). Catalyst CAT−2019 was provided by College of Advanced Materials Engineering, Jiaxing Nanhu University (Jiaxing, China). Phosphate−buffered solution (0.01 mol·L^−1^, pH = 2, pH = 7) was supplied from Xiamen Amemani Biotechnology Co., Ltd. (Xiamen, China). Phosphate−buffered solution (0.01 mol·L^−1^, pH = 12) was bought from Yida Technology (Quanzhou) Co., Ltd. (Quanzhou, China). Triacylglycerol lipase (≥700 unit/mg solid) was obtained from Sigma−Alrdich. Ltd. (Shanghai, China).

### 3.2. Synthesis of PBATGA Copolyesters

PBATGA copolyesters were synthesized by one−step direct melt polycondensation. The synthesis process was divided into two stages: esterification and polycondensation. The specific synthesis route is shown in Figure 1.

(1)BDO, TPA, AA, MG and CAT−2019 were added into a 100 mL three−flask for reaction, the molar ratio of alcohol and acid was 2.0:1.0, the molar ratios of PTA+AA and MG were 100:0, 90:10, 80:20, 70:30 and 60:40. The esterification reaction was conducted from 200 to 230 °C for 4~5 h under N_2_ atmosphere until the water yield reached 95% of the theoretical value. Among them, the molar ratios of TPA and AA were 50:50, and the molar ratios of SA and GA were 100:0, 90:10, 80:20, 70:30, 60:40 and 50:50. The corresponding products were named from PBATGA10 to PBATGA40.(2)The condenser tube and water separator were removed, then the pressure was slowly decreased to less than 20 Pa and the temperature gradually increased to 230 °C and continued for about 4~5 h. During this period, the mixing speed was slowly reduced to 160~60 r/min according to the change of viscosity. When the viscosity of the product was high or the phenomenon of climbing rod occurred, it was discharged for use. Finally, the obtained white to faint yellow products could be used directly in subsequent tests without further purification. The appearance of the final product is shown in Figure 7. PBAT appears bright milky white, while PBATGA copolyesters appeared light yellow.

### 3.3. Characterization

(1)The molecular weight and the molecular weight distribution were investigated using Gel Permeation Chromatography (GPC) on an Acquity APC advanced polymer chromatography system advanced performance gel permeation chromatograph, and the mobile phase was tetrahydrofuran.(2)The chemical structure was recorded using ^1^H NMR on a Bruker Avance NEO 500 MHz apparatus. Deuterated chloroform was the solvent for testing.(3)The characteristic viscosity test was performed as follows. First, a sample of about 25 mg was weighed and phenol−tetrachloroethane 1:1 mixed solution was used as a solvent to set the volume in a 25 mL volumetric flask. Under the condition of 30 °C, the outflow time t_0_ and t of pure solvent and polymers were tested by a three−tube viscometer with a pipe diameter of 0.7~0.8 mm. Each sample was measured three times in parallel, and the average value was obtained, which is the outflow time of the solution to be measured. The formula for calculating the characteristic viscosity is the Solomon–Ciuta method:
(3)$$\eta_{r}=\frac{t}{t_{0}}$$
(4)$$\eta_{\mathrm{sp}}={\;\eta }_{r}-1$$
(5)$$\;\left[\eta \right]=\frac{{\left[2\left({\;\eta }_{\mathrm{sp}}-\ln\eta_{r}\right)\right]}^{\frac{1}{2}}}{C}$$
where η_r_ is the relative viscosity, η_sp_ is the specific viscosity, C is the concentration of the solution, t_0_ is the time when the pure solvent flows out of the viscometer (unit: s) and t is the time when the polymer solution flows out of the viscometer (unit: s).
(4)Differential scanning calorimetry (DSC) analysis was performed on a DSC 2500 instrument (TA Company, Boston, MA, USA). A sample with a mass of 10 mg was taken and put into an aluminum pan. The samples were first heated to 200 °C at 10 °C·min^−1^, kept at 200 °C for 3 min, and then they were cooled to −50 °C at 10 °C·min^−1^ and kept for 3 min. They were then reheated again to 200 °C at the same heating rate.(5)Dynamic Thermomechanical Analysis (DMA) was obtained using the DMA 8000 dynamic mechanics analyzer of Perkin Elmer (Waltham, MA, USA). The oscillation frequency was 1.0 Hz, the amplitude was 50 μm, the specimen size was 1.0 mm × 8 mm × 20 mm and the tensile mode test was performed. The test temperature range was −80~100 °C, and the heating rate was 5 °C·min^−1^. The peak value of the loss factor over this temperature range was the glass transition temperature T_g_.(6)Thermogravimetric analysis (TGA) was measured using the TGA7 (Thermogravimetric Analyzer) type thermogravimetric analyzer of PerkinElmer (Waltham, MA, USA). The test temperature was 40–800 °C with a heating rate of 20 °C·min^−1^ under N_2_ atmosphere.(7)For the mechanical properties test, the samples were prepared according to the GB/T1040–1BA type, and the tensile test was conducted by using universal testing machine at room temperature at the tensile speed of 20 mm/min. The tensile strength and elongation at break were calculated and recorded.(8)Degradation experiments of PBATGAs samples (0.4~0.5 mm in thickness) were carried out in four buffer solutions of acidic (pH = 2), neutral (pH = 7), alkaline (pH = 12) and 0.2 mg /mL of Lipase in phosphate−buffered solution. All the samples were kept at 37 °C, the buffer solutions were replaced and the samples were taken every 7 days. After cleaning, the samples were dried at 60 °C for 24 h. The degradation behavior was evaluated by the weight percentage of residue, according to the following Equation (6):
(6)$$\;\mathrm{Residue}\;\mathrm{Weight}\;\%=\frac{W_{t}}{W_{0}}\times 100\%$$
where W_t_ is the weight before degradation, g; W_0_ is the weight after degradation, g.


## 4. Conclusions

We successfully utilized methyl hydroxyglycolate to prepare a series of PBATGA copolymers with GA units ranging from 0% to 40% through quaternary copolymerization. The main conclusions are summarized as follows:(1)The results of DSC test revealed that the PBATGAs were semicrystalline polyesters with T_m_ > 100 °C when the content of GA units <40%.(2)The results of TGA and tensile test revealed that the PBATGA copolymers possess good thermal and excellent mechanical properties. The tensile strength was 5.98~18.58 MPa, and the elongation at break was more than 463%. In particular, the elongation at break of PBATGA40 was even more than 1200%. These results were better than the most degradable polymers used as membranes.(3)The results of DMA test revealed that the T_g_ values of PBATGA10–40 were between −20.1 °C and −14.5 °C, which are close to the T_g_ of PBAT (−17.4 °C).(4)Regarding the hydrolysis experiment in four kinds of buffer solutions, the weight loss of PBATGA copolymer in descending order, is as follows: pH = 12 > Lipase ≈ pH = 7 > pH = 2. In the alkaline solution with pH = 12, the weight loss of PBATGA40 copolyester was 58.29% after 49 days, and only 12.67% and 17.21% in acidic and neutral solutions, respectively. Lipase could not effectively catalyze the hydrolysis of the ester bonds in PBATGA copolymers.(5)GA units can accelerate the degradation of PBAT in the presence of water. Therefore, the above experiments proved that the PBATGA copolyesters are promising candidates for conventional nondegradable plastic products in the field of direct environmental leakage. In the future, efforts to study the shelf life of PBATGA copolyesters in vacuum are very necessary.

## Data Availability

The authors declare that the data supporting the findings of this study are provided in the main article and can be accessed upon request via email to the corresponding authors.

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
