# Peer review of "Study on Properties and Degradation Behavior of Poly (Adipic Acid/Butylene Terephthalate-Co-Glycolic Acid) Copolyester Synthesized by Quaternary Copolymerization"

_ijms, 2023, doi:10.3390/ijms24076451_

Round 1

Reviewer 1 Report

The paper's subject is interesting. Generally, this article is well organized but needs improvement in the presentation and discussion of the results.

·         The paper's contribution needs to be clarified. Please improve de discussion about your polymerization process, and the present work should be compared with the results in the literature to highlight its novelty.

·         In Table 2, is it consistent with using decimals numbers on temperature? What is the experimental error of the equipment?

·         Why did Tc, Tm, and Xc decrease with the increase in GA quantity? The discussion of these results needs to be improved.

·         Please present the results of Young's Modulus and Yield Stress of the stress-strain curve and discuss them;

·         It is also better to present the Mechanical properties in a Table with the standart desvion.

·         The presence of GA reduced the strain-hardening stage in the stress-strain curve. Discuss this warp mechanism change.

·         In the abstract, the authors state that adding GA units has a slight effect on the shelf life of the copolymer. However, only molar mass analysis over time was carried out. For example, it is impossible to infer its behavior concerning mechanical properties.

·         Pay attention to the statistical treatment of data. Add the number of replicates for each experiment and each analysis and report data with standard deviations/error bars.

Reviewer 2 Report

The authors present a study on properties and degradation behavior of poly (adipic acid/butylene terephthalate-co-glycolic acid) co-polyester synthesized by quaternary copolymerization. Four kinds of PBATGA co-polyesters were prepared and their molecular weight, molecular structure, crystallinity, thermal and mechanical properties were analyzed. In addition, a study of the hydrolysis behavior of these co-polyesters was done in different media for 49 days by testing the weight loss rate of the materials.

In my opinion, some issues must be clarified before further consideration:

-        Extensive English syntax correction;

-        Too long Introduction section;

-        All abbreviations, including characterization methods should be named at their first use (DSC for example);

-        Number the equations coherently. Equation (6) appears first, then the others from 1 to 5 in the text;

-        Description of the measures that appear in equation (6) is a little bit confused;

-        Graphics in Fig. 6 are not uniformly depicted (Fig.6b differs from the others);

-        A discussion based on the results obtained by each method of analysis must be made considering PBAT and other similar degradable polymers;

-        Define C in the equation (3) for characteristic viscosity calculation/page 13, range 333

-        Please discuss and conclude about the possibilities of use of these copolymers in accordance with the results obtained.

Round 2

Reviewer 1 Report

The authors made all the suggestions

Author Response

Dear reviewer:

Thank you for your constructive comments.

Reviewer 2 Report

The authors answered to the suggestions from the first stage of the evaluation. However, there are some minor points that must be resolved before moving on:

-At page 5/17: “The crystallinities (Xc) were calculated according to Equation (2)…

Where ΔHm0 is the melting enthalpy of 100% crystallized PBAT is the enthalpy of 114 J/g[34].”

This statement makes no sense. I suggest to carefully read the reference [34] and explain correctly this equation.

-Page 8/17: “The young's modulus…” – Please correct: “Young's…”

-Page 13/17:(6) Thermogravimetric analysis (TGA)…please correct

-In the Conclusion section: “We successfully…”

-And please reformulate the following paragraph (the last in the Conclusion section): “GA units can accelerate the degradation of PBAT in the presence of water, therefore. The above experiments proved that the PBATGA copolyesters are promising candidates for conventional nondegradable plastic products in the field of direct environmental leakage and .”
